# Living the employer brand during a crisis? A qualitative study on internal employer branding in times of the COVID-19 pandemic

**Marthe Rys** * , **Eveline Schollaert** , **Greet Van Hoye**

Department of Marketing, Innovation and Organisation, Ghent University, Ghent, Belgium

These authors contributed equally to this work.
* marthe.rys@ugent.be

## Abstract

Employer branding has emerged as a strategic imperative in the quest for talent. However, existing research has predominantly explored stable periods, overlooking the possible transformative impact of crises and the crucial role that HR managers play in crafting internal employer branding strategies. As such, this research addresses this by scrutinizing internal employer branding during the COVID-19 pandemic. Conducting in-depth interviews with 37 Belgian HR managers, we delve into the perceived challenges and opportunities that the COVID-19 crisis presented with respect to internal employer branding and its touchpoints—internal communication and leadership. A subsequent member and employee check with six HR managers and six employees validated our findings. The results unveiled organizations' heightened concern for employer branding during crises, emphasizing the strategic reflection invested. Remarkably, despite facing organizational/operational constraints/risks imposed by the crisis, the attention and efforts remain steadfastly centered on the experienced internal employer brand in crisis situations. Additionally, a contextual analysis suggests that various employer brand types face similar challenges in crises, however, the employer brand serves as a defining factor that shapes how an organization responds to both external uncertainties and internal dynamics brought about by the crisis. This study contributes to a nuanced understanding of internal employer branding dynamics during crises, shedding light on the strategic considerations of HR managers.

## Introduction

Since the expression 'war for talent' suggested by McKinsey, organizations are applying different approaches to compete for talent. Literature has proposed employer branding as an effective organizational strategy to win the 'war' by attracting and retaining talented employees [1]. Employer branding aims to differentiate the organization, internally and externally, through its unique employment experience and attractiveness as an employer [2,3]. To enhance attractiveness, creating and promoting a strong employer within the organization (i.e. internal

**Competing interests:** The authors have declared that no competing interests exist.

employer branding) and externally promoting the employer brand through recruitment strategies (i.e. external employer branding) are crucial elements.

Research has extensively studied the external promotion of the employer image. Internal employer branding, however, has received little attention compared to external employer branding [4]. Backhaus and Tikoo [5] described internal employer branding as a strategy that creates, promotes, and delivers a distinctive and attractive image as an employer (i.e., employer brand) to employees. To achieve an effective internal employer branding approach, the employer brand has to be present in each aspect of the organization [6]. Hence, employees should experience the internal employer brand through multiple touchpoints, such as internal communication and leadership [7,8]. These aspects, already established as important during stable times, form integral components of an effective internal employer branding strategy. Prior research suggests that a strong internal employer brand relates positively to employees' attitudes and reduces turnover intentions [9,10]. However, previous studies typically focused on employee perceptions, largely ignoring the crucial role that HR managers or employer brand managers play in crafting and implementing internal employer branding strategies [11]. Hence, we know little on how strategic decisions regarding internal employer branding are made and implemented.

Furthermore, to extend our knowledge of internal employer branding, scholars like Lievens and Slaughter [12] advocate for exploring the employer brand amid unstable times, emphasizing that current literature predominantly addresses stable environments. This imperative gains significance in an era where crises, fueled by factors such as the rise of social media and accelerated media flows, occur more frequently, amplifying threats to reputations [13,14]. During a crisis, the employer brand can be vulnerable to damage and negative influences [15]. Crises intensify the pressure on organizations to make strategic decisions. These decisions impacts the immediate well-being of employees that experience enhanced levels of frustration, uncertainty and need for information during a crisis [16]. Additionally, during crises, strategic decisions can influence employees' perceptions of the employer brand, which may prompt questions about the alignment between the organization's stated employer brand values and the actions taken during challenging circumstances. For instance, if an organization claims to be a warm and supportive employer, will this promise hold on when things get tough and lay-offs may seem necessary? Moreover, is the employer value proposition still relevant during a crisis or do certain other aspects become more important, such as creating more togetherness among employees or focusing on the operational functioning? Furthermore, some employer branding practices might become difficult or even impossible during a crisis, when resources might be lacking or social media unavailable. Navigating this delicate balance between addressing immediate operational needs and preserving the long-term reputation of the employer brand might pose significant challenges for organizations and HR managers are likely to play a crucial role in shaping the internal employer brand during such transformative events.

The COVID-19 pandemic, with its unprecedented disruptive nature and its impact on employees' careers and well-being [17,18] presents a unique opportunity to delve into the intricacies of internal employer branding during a crisis. This crisis not only necessitated organizational adaptations but also left an indelible impact on employees [19,20]. During the pandemic, organizations were forced to adopt new working routines and policies to ensure business continuity. At the same time, these restrictions and working routines threatened employees' well-being due to the inability to separate work-private life, the feeling of job insecurity, and loneliness [17,18,21]. It created an environment where employees experienced uncertainties, changing work patterns, and feelings of isolation. Thus, the COVID-19 pandemic, with its challenges, has been a disruption that overturned existing working arrangements [22] and threatened employees' well-being [23]. Therefore, examining internal

employer branding during the COVID-19 pandemic provides a suitable context to understand how HR managers navigate challenges and safeguard their organization's employer brand in times of crises.

This study aims to deepen our understanding of how internal employer branding is influenced by disruptive events, specifically exploring the strategic decisions made and implemented by HR managers during the pandemic. We also aim to investigate the role of the key internal employer branding touchpoints internal communication and leadership during a crisis, as they may be vital for adapting employees to new conditions and challenges [24]. For instance, the provision of emotional support, empathy, a relationship-oriented communication, and supportive leadership might enhance a sense of togetherness, trust, and support among employees, potentially fostering commitment to the employer brand during the pandemic.

To this end, in-depth interviews were conducted with 37 HR managers from 37 Belgian organizations that varied in sector and size. To validate our findings, we conducted a member check with six other HR managers from different organizations and an employee check with six employees from the organizations that participated in the study. On a theoretical level, our study makes at least two significant contributions. First, we shed light on how HR or employer brand managers make and implement strategic decisions regarding internal employer branding. We thus extend prior research that has mostly looked at employer brand perceptions held by employees and answer the call for more organization-level, strategic HRM research. Second, our research explores the dynamics of internal employer branding during crises, offering insights into the challenges and opportunities they may create and how HR managers respond to them. These insights complement prior research conducted in stable times. Regarding practical implications, our findings are useful for HR managers wanting to learn from their colleagues and how they have handled internal employer branding during disruptive events. It can make them more aware of the strategic considerations involved and their own decision-making process.

## Background

### Employer branding

Lievens et al., [25] described employer branding as the promotion of a clear view of what makes the organization different and desirable as an employer to internal and external audiences. According to Ambler and Barrow [26], this image can be described as "the package of functional, economic, and psychological benefits provided by employment, and identified with the employing company" [26, p.187]. It consists of both instrumental (e.g., pay, career opportunities) and symbolic attributes (e.g., warm, innovative) associated with an employer [27].

Employer branding is a strategic function that communicates organizational values to its stakeholders. It provides the opportunity for employees to consider if the organization fits their values [28,29]. Ultimately, employer branding helps realizing that the best qualified employees enter and remain in the organization. Hence, employer branding is a long-term strategy that intends to attract and retain current and potential employees by promoting a distinctive and attractive employer image to these audiences [30].

Promoting an employer brand externally can establish the firm as an "employer of choice" and enables the organization to attract the best candidates. However, a strong employer brand also has a significant impact on employee attitudes [10]. It improves key outcomes, such as organizational performance [31], applicant attraction, recruitment efficiency, employees' engagement and commitment [30], and reduces turnover intention [9].

Despite the positive impact of internal employer branding on the organization, the main focus of employer branding literature has traditionally been on external employer branding [32]. Literature has primarily shed light on which attributes applicants consider as attractive in an employer [e.g. 27] and the impact of employer brands on early-stage applicant outcomes [e.g. 33]. As such, our knowledge of how organizations approach their internal employer branding among current employees is limited [34]. Therefore, we specifically explore internal employer branding during unstable times to further extend empirical knowledge.

## Internal employer branding

Internal employer branding is the promotion of what makes the organization different and desirable as an employer to an internal audience [25,35]. Originally, internal employer branding stems from a marketing concept, called internal branding. According to Barros-Arrieta and García-Cali [36], internal branding is "a strategy to promote the brand within the organization to ensure that employees adequately deliver the brand promise to external customers" [36, p. 136]. Moreover, internal branding aims to affect customers' experiences and perceptions of the brand through employee knowledge, commitment, and behavior [37,38]. While internal branding promotes a brand, product, or service among employees and intends to motivate them to communicate this to external customers, internal *employer* branding promotes and radiates an image of what makes the employer unique and desirable among current employees. Therefore, internal employer brands describe how internal audiences perceive the organization's employment offer, unique work experiences, and employer values.

Scholars have proposed that the employer brand should emerge in every part of the employment experience [6]. In every operational and interpersonal touchpoint with employees, the employer brand should be clearly present and consistently managed [39]. Particularly, internal communication and leadership are important touchpoints to transfer the employer brand to employees. Staniec and Kalińska-Kula [40] suggested that these are essential touchpoints to facilitate the delivery of the employer brand. However, so far limited empirical research has examined the role of internal communication and leadership in internal employer branding. Moreover, in literature there is no insight yet in how these touchpoints might be altered by HR managers during a disruptive event to align with the requirements of the employer branding strategy.

**Internal communication.**   Internal communication, as defined by Kalla [41], encompasses all formal and informal exchanges of information within an organization, playing a crucial role in daily and strategic activities [42]. Recognized as pivotal, it is fundamental for successful employer brand development and management, ensuring consistent perceptions among internal and external stakeholders [43]. Internal communication about the employer brand, managed by HR managers, serves as a tool for conveying employment advantages, sharing core values, aligning employee behavior with the employer brand, and fostering commitment and loyalty [6,44–46].

Moreover, it aids in developing committed and loyal employees who identify with the employer brand values [36]. According to Itam et al., [47], internal communication reduces information asymmetry among current employees by sharing the organization's mission and values. HR managers view internal communication as collaborative employer brand crafting [48], facilitating the exchange of information and unique values related to the employer brand [42]. It plays a pivotal role in shaping employee attitudes toward employer branding [7,37]. Effective communication is thus a key factor in successful employer brand development and management, ensuring uniform perceptions among internal and external audiences and employees' attitudes toward the employer brand during stable times.

In times of crisis, organizations face the challenge of reassessing traditional approaches that may prove inadequate under the pressures of such situations. The COVID-19 pandemic introduces new working practices, particularly the transition to remote and hybrid work, potentially influencing how employees perceive the employer brand. Recognizing the need for a deeper understanding, organizations may consider adopting a communication strategy focused on fostering relationships, unity, and a sense of togetherness, aiming to ensure employee satisfaction during a crisis. Nelke [49] shows that this can help to increase trust among employees. Moreover, according to Sinčić Ćorić and Špoljarić [42], satisfaction with internal communication is important for employees to foster engagement and employee satisfaction. Therefore, to craft an effective internal communication strategy crucial for shaping, understanding, and fostering a positive perception of the employer brand during a disruptive event, HR managers may need to adapt the delivery approaches, modify content, or alter the manner in which it is presented to employees.

**Leadership.** Supervisors can influence, motivate, and enable employees to achieve a specific goal [50]. Leadership concerning employer branding includes all the efforts of leaders to disseminate the employer brand principles to employees, which entails showing the work culture and unique work practices. HR managers rely on supervisors to convey the employer brand to employees, forming a crucial partnership in delivering the employer brand to employees. Zeesahn et al., [51] emphasized that organizations should focus on both leadership and internal employer branding to retain and attract employees. Moreover, leadership has a significant part in creating an employer brand climate and establishing an inner atmosphere which impacts employees' attitudes [52].

Barrow and Mosley [53] emphasize supervisors' pivotal role in developing and communicating the employer brand. For instance, supervisors can actively embody the employer brand by sharing personal experiences that reflect the employer brand values during team meetings. Concurrently, supervisors, as suggested by Biswas and Suar [54], can provide clear directives, like implementing regular team-building activities, to showcase distinctive work practices. These visible leadership actions not only influence how employees perceive the internal employer brand, values, and culture, but can also support them in incorporating these aspects into their daily tasks [54]. For instance, during a daily team meeting, employees can proactively contribute to a warm and collaborative atmosphere by sharing positive experiences related to teamwork or recognizing a colleague's effort. This daily practice aligns with the employer brand's emphasis on collaboration and fosters a positive workplace culture. Therefore, to achieve a strong internal employer brand, supervisors have to actively "live the employer brand" and display the employer brand in their interactions with employees.

Nevertheless, during a disruptive event, supervisors can rise to the challenge while other supervisors will struggle to manage the crisis [55]. Supervisors are relying on their instincts and insights, supported by HR managers, to ensure their employees feel supported. Dirani et al., [55] concluded that when employees face challenges in a new work environment, supervisors must become more flexible to adjust their employees to the new altered work environment. Moreover, supervisors may need to implement targeted approaches to address the likely impact of a crisis on employees' motivation, satisfaction, and overall well-being. This could involve strategies such as transparent communication, empathetic leadership, and tailored support mechanisms [56]. As such, to foster employer brand-aligned leadership during turbulent times, it is crucial to explore how HR managers guide supervisors in navigating challenges and promoting the employer brand among employees.

## Aims of the study

Amidst the dynamic landscape of disruptive events, marked by an anticipated increase in the frequency of crises [21], there exists a noticeable gap in the exploration of challenges and opportunities pertaining to internal employer branding and the touchpoints internal communication and leadership. The existing body of research has predominantly focused on stable times and employee perceptions. In response to the call of Lievens and Slaughter [12] to investigate employer branding in unstable times, we explore how HR managers make and implement strategic decisions regarding internal employer branding and its associated touchpoints during a crisis. How HR managers approach the intricacies of internal employer branding during tumultuous times, and the inherent difficulty of adhering to established employer brand values amidst crisis-induced shifts, remains insufficiently understood. The uncertainty and pressure of a crisis may necessitate strategic decisions that could potentially deviate from the conventional employer branding focus. Questions arise about the feasibility of upholding a warm and supportive employer image during times of organizational upheaval, such as layoffs or operational adjustments. Hence, the exploration of internal employer branding becomes imperative, offering insights into the nuanced strategies employed by HR managers and the potential shifts in focus within employer branding practices.

Therefore, it is our goal to provide insight into how HR managers perceived and experienced the impact of the COVID-19 pandemic on internal employer branding and the two touchpoints internal communication and leadership. As the pandemic yielded negative implications for organizations' business continuity, we can expect a similar challenging impact on internal employer branding and touchpoints internal communication and leadership. Therefore, the following research question was formed:

*Research Question 1: What challenges have HR managers experienced during the COVID–19 pandemic regarding internal employer branding and internal employer branding touchpoints (i.e. internal communication and leadership)?*

Furthermore, in times of crisis, managers have an exceptional opportunity to learn or refine strategic decisions, which may have not been straightforward before. According to Akkermans et al., [17], employees or organizations who experience a critical negative workplace event are more likely to engage in improvisation behaviors such as taking advantage of opportunities and "thinking outside the box" [57]. Dealing with these challenges, HR managers might take a different and more creative employer branding approach to address changing employee needs and concerns. Accordingly, we can expect that possible opportunities may arise during critical events for internal employer branding and its touchpoints. As such, the following research question was formed:

*Research Question 2*: *What opportunities have HR managers experienced during the* COVID–19 *pandemic regarding internal employer branding and internal employer branding touchpoints (i.e. internal communication and leadership)*?

## Materials and methods

As we aim to provide insight into how HR managers perceived and dealt with the COVID-19 pandemic, it is important to describe the COVID-19 context and delineate what is unique during the data collection.

## The COVID-19 context

The data were collected during the second wave of the pandemic in Belgium, from December 2020 until April 2021. The government took strong measures to limit the outbreak, which had consequences for both employers and employees. First, telework was mandatory for all companies, unless it was needed for the continuity of the products or services. If employees were allowed to work on-site, social distancing and wearing face masks were the norm. Lastly, the hospitality sector (e.g. restaurants, hotels) and non-essential stores were obligated to close. We excluded these organizations from our sample, as we only wanted to include organizations that were operating during the pandemic.

## Ethical approval and consent

Our study adhered to the stringent ethical protocol of the authors' university. Therefore, obtaining specific ethical approval was not required, which was confirmed by the university's Committee Ethical Affairs Faculty of Economics and Business Administration (Ghent University). Participants from the interviews, member check, and employee check provided both written and verbal informed consent, involving a detailed explanation of the study's objectives, purposes, and procedures. Participants were informed about key aspects, including the voluntary nature of their involvement, the option to withdraw at any point, the recording process with privacy safeguards, permission for data processing, and the opportunity to access the research findings. Only the first author has access to the personal information of the individual participants. Anonymity and confidentiality measures were rigorously implemented, employing code numbers, a distinct file linking personal information to these code numbers, anonymized data files, restricted access to data, and safeguards to protect any identifiable elements.

## Data collection

**Sample.**   The sample consisted of 37 Belgian HR managers from 37 different organizations. We interviewed participants that were responsible for the employer branding approach in their organization. Of the 37 organizations, 12 (32%) were small- and medium-sized and had less than 250 employees, and 25 (68%) were large organizations employing more than 250 employees. Two organizations are situated in the public sector, whereas the others are in the private sector, such as IT, consultancy, and insurance. Concerning the participants, managers were between 25 and 62 years old ($M = 45.24$, $SD = 9.59$), and 58% were female. Their experience ranged from several months to 31 years ($M = 13.33$ years, $SD = 9.60$). See 'S1 Table' for more detailed information about the participants.

**In-depth interviews.**   As the perspective of HR managers about internal employer branding during unstable times lacks clear understanding and knowledge [11,12], a qualitative approach was adopted. Qualitative research—due to its innate explorative nature—allows for more detailed insights into a topic, as it builds upon the perceived realities of its target population [58]. It enables us to capture the nuances of this relatively unexplored topic. Therefore, this study specifically delved into the perceptions and experiences of HR managers regarding their internal employer branding strategies and practices during the pandemic.

Our participant selection involved purposeful and maximum variation sampling [59]. Purposeful sampling was utilized to identify and choose participants whose characteristics align with the focus of our investigation [60]. It is particularly useful when the researcher intends to recruit participants who possess specific information based on their professional experiences [61,62]. For the current study, we intended to recruit participants that were responsible for (internal) employer branding. Furthermore, since the objective was to enhance our comprehension of the phenomenon [63,64], we intentionally sought diversity through maximum

variation. Consequently, we selected organizations that maximally differed on two conditions that may influence employer branding activities, namely organization size and sector [65,66].

To facilitate open sharing of experiences and perceptions, including potential negative views or personal accounts, we chose individual semi-structured interviews to collect the data [67,68]. The interviews were carried out by two master students in HRM and a doctoral researcher. We trained students' interview techniques before and during the interviews. Using our network, we contacted organizations via email in which we specified the topics to be addressed. The interviews were conducted through video calls, as teleworking was the norm at that time, and were recorded.

The interviews were based on insights retrieved from a literature review. Before we started the interview, we made sure that there was a common understanding of the concept of internal employer branding by asking what they understood regarding this topic and providing a definition of this concept. The interview guide covered three main categories: (a) internal employer branding, (b) internal communication about employer branding, and (c) leadership in function of employer branding (see 'S2 Table'). Furthermore, in line with guidelines, the initial interview questions were regularly updated based on the presented information by HR managers during this data-collection process [69]. We applied the principles of data saturation and information redundancy. Our data collection ended when additional interviews did not bring new information. This was achieved after about 30 interviews, but seven control interviews were still conducted to be sure. Participants were instructed to provide honest answers based on their experiences during the COVID-19 pandemic.

**Member check.**    We conducted a member check in April 2022 to validate and verify the research findings obtained from the interviews [70]. We contacted ten (other) Belgian HR managers through email and received answers from six of them. We summarized our results and compiled them into 16 statements and asked whether the HR manager experienced something similar in their organization during the pandemic. Before we presented these statements to the participants, we defined internal employer branding. Participants had the opportunity to elaborate on their choice. None of the 16 statements was fully rejected by the participants, increasing our confidence in the validity of the findings. Consult 'S1 Table' for participant details.

**Employee check.**    Finally, we conducted employee checks to ensure that the conclusions derived from the interviews were experiences that were recognizable to the employee target group [71]. We extend our gratitude to the reviewer for the valuable recommendation to include employee interviews in our research. This allowed us to verify whether members of the target group of internal employer branding agreed with our findings. Moreover, this enabled us to supplement our results with their input. As such, in December 2023, we carried out an employee check to authenticate and confirm the research findings from the employees' standpoint. We reached out to all 37 organizations and conducted interviews with six employees, each lasting between 25 and 35 minutes. During these semi-structured interviews, we inquired the participants about their experiences with the strategies and practices related to employer branding that were implemented during the COVID-19 pandemic, using the 16 statements and results as a basis for discussion. Prior to posing these questions, we ensured that employees had a clear understanding of the concept of employer branding. Subsequently, employees were given the opportunity to elaborate on these statements. Consult 'S1 Table' for participant details.

## Data analysis

We implemented the thematic analysis of Braun and Clarke [72] to discover patterns in the data. Our research design has both an inductive and deductive component, therefore, thematic

analysis is the most appropriate approach. The interviews were structured around themes identified in prior literature [73]. We were receptive to investigating various subthemes associated with internal employer branding and touchpoints internal communication and leadership, and the emergence of new themes. See supplementary information in 'S2 Table' for insights into data analysis and 'S3 Table' for an overview and description of the (sub)themes.

The thematic analysis consists of six phases and was performed using NVivo 8. In the first phase, we familiarized ourselves with our data by transcribing and (re-)reading the data. During this step, we marked sentences and wrote down our first impressions. In the second phase, prior to initiating the coding process, we pre-established nine themes deductively, including categories such as "policy," "policy challenges," and "policy opportunities". Subsequently, initial codes were generated based on patterns that emerged from the collected data. During the initial coding, we were open to new insights and contradictions and considered the organizational context. The first author was mainly responsible for the coding, but the coding and themes were frequently discussed with the other authors, and issues were resolved in consensus. In the third phase, following the coding of 15 interviews, our initial focus involved identifying new subthemes within our coding. Themes like "loss of contact with the employer," "digitalization", and "new leadership skills" surfaced during this phase. Upon completing the coding for all interviews, we conducted an analysis by shifting our attention to a broader level of themes. This involved identifying overarching themes, such as various types of employer brands, organizational size, and sector. We built a thematic map of the codes and reflected on the relationship between these themes. At the end of this step, we identified three main themes (e.g. "internal employer branding", "internal communication", and "leadership") and six themes ("policy challenges", "policy opportunities", "internal communication challenges", "internal communication opportunities", "leadership challenges", and "leadership opportunities). As thematic analyses allow for the emergence of inductive themes, we identified a total of 24 inductive subthemes within the deductive (main)themes. These included topics like "flexibility and well-being," "retention and strengthening of employer brand values," "loss of informal communication and connectedness," "message," and more. In the fourth step, we reviewed, refined, and renamed the themes and looked for overlapping themes. While retaining the three main themes and six themes, we settled on 23 subthemes. Notably, we discussed and opted to merge the themes "leadership visibility" and "fostering communication" into a single theme, named "Leadership visibility through communication". Next, in the fifth step, we assessed intercoder reliability using the Krippendorf's alpha (Kalpha) statistic, a commonly employed measure for such evaluations [74]. A random subset of quotes was provided to another team member. Utilizing the coding reference framework, which includes detailed content information for the subthemes (see 'S2 Table'), the team member assigned codes to the subthemes following the method outlined by O'Connor and Joffe [74]. This process yielded a Kalpha value of .81, signifying substantial agreement. Ongoing discussions and assessments ensured the continued robustness of our analysis. We made the decision to merge 23 subthemes into 16 subthemes, for instance "Reinforcement of internal employer branding" and "Retention and strengthening of employer brand" consolidating them into a subtheme renamed as "Continuous focus on the internal employer brand". In the sixth step, we implemented the subthemes in the results section and elaborated on these subthemes by providing evidence (i.e. quotes from participants) (see 'S4 Table'), conducting a member and employee check.

## Results

In this section, we discuss how HR managers experienced challenges and opportunities during the pandemic related to internal employer branding and the touchpoints. We did not identify

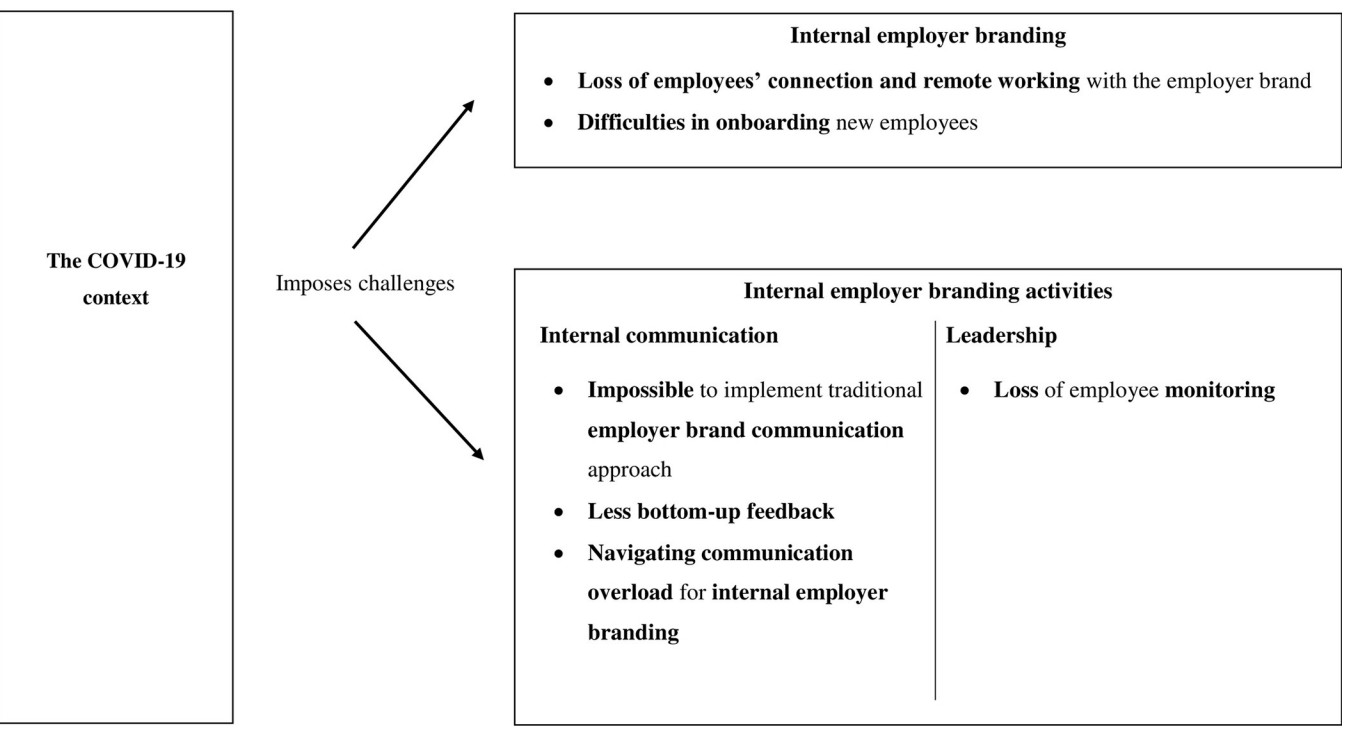

**Fig 1. A research model on the perceived influence of the Covid–19 pandemic on internal employer branding.**

significant variations concerning organization size or sector. However, distinctions emerged in the approach of HR managers during the COVID-19 pandemic based on their employer brand values or employer value proposition. Fig 1 provides an overview of our findings.

## Main theme 1: Internal employer branding

**Theme 1.a: COVID-19 imposes challenges.** *Subtheme 1.a.1*: *Loss of connection and remote working*. HR managers perceived that the COVID-19 pandemic influenced their internal employer branding in a restrictive and challenging manner. One HR manager reflected on this, stating: "*It was new for everyone; no one knew how to handle it. . . managing the employer brand during the COVID-19 pandemic was a new experience for us*" (female, large organization, medicine). First, HR managers mentioned experiencing less employee connectivity and commitment to the internal employer brand. A vast majority of HR managers identified that employees had difficulties in recognizing the employer brand. An HR manager reported:

> That is what you try to do as an organization, to maintain the connection with the employer brand in times of COVID. I am convinced, but that is personal, that physical contact, being together in the office, is still much better than remotely. You can see when something is going on with someone. Now, if something is going on with someone, they will call me, but there is still a certain restraint, a certain distance. I sometimes worry about people too. If you do not feel well, had an argument at home, and you come into the office, well, just sit down for a moment. I was always the first one in the office. When HR people came to say hello, I could see if something was wrong, and we would sit down for a coffee and discuss what was going on. We do not do that anymore, so that human care of our employer band has somewhat disappeared. (female, small organization, information technology)

Moreover, most HR managers identified the mental strain on employees due to fears of COVID-19 and challenging home situations. They acknowledged the challenge of restoring a positive drive among employees. These experiences underscore the intricate interplay between flexibility, well-being, and internal employer branding, highlighting the multifaceted strategies HR managers had to employ to navigate these challenges while maintaining a positive work environment aligned with the employer brand.

*Subtheme 1.a.2*: *Difficulties with onboarding new employees*. Second, due to physical distance, HR managers perceived more difficulties to get new employees involved with the employer brand of the organization. Moreover, some HR managers mentioned that the conventional onboarding practices aimed at familiarizing employees with the employer brand, including components such as welcome packages, tours, and personal introductions, have faced significant challenges due to the constraints imposed by remote work. One participant elaborated:

> But for example, we hired someone in October, and typically, when someone joins us, there is a welcome package waiting, a guided tour of the company, on–the–job training, introductions to the team, and we organize a lunch. None of that has happened now, so that person came to the office for just one day to meet with the manager, but then the rest of the time, they have been working from home. I regularly call them to ask how things are going. We feel somewhat guilty because they are all alone there. They mentioned that it is quite challenging integrating into an organization without knowing the ropes and employer brand. (male, small organization, information technology)

**Subtheme 1b: COVID-19 creates opportunities.** *Subtheme 1.b.1*: *Reflecting about the internal employer brand*. Although COVID-19 imposed challenges for organizations, respondents also articulated how they experienced the pandemic positively. Despite these challenges imposed by the crisis context, employers mentioned that they reflected on the internal employer brand. Some organizations redeveloped their internal employer branding or reflected on how to improve this. According to three HR managers that were interviewed, the COVID-19 pandemic created a context that aided in accelerating or refocusing internal employer branding. Other respondents mentioned that the COVID-19 pandemic did not force them to change the content of the internal employer brand, but supported them to reconsider the delivery of the internal employer brand, as one HR manager elaborated:

> In terms of content, the employer brand has not changed, but it has been implemented and transferred faster and was repeated more often. Because employees work from home, we had to trust employees to radiate the employer brand correctly, because we cannot know how employees implement the employer brand. In addition, COVID–19 created the opportunity to transfer and position the employer brand differently. The circumstances of the pandemic forced us to do this. (male, large organization, social sciences)

*Subtheme 1.b.2*: *Continuous focus on the internal employer brand*. Despite these challenges, a vast majority of the respondents reported that they continued to focus on internal employer branding. Specifically, one HR manager highlighted that the COVID-19 pandemic underscored the need to bolster their retention strategy through the effective management of internal employer branding during the pandemic. Furthermore, a significant number of HR managers mentioned to adapt a flexible approach in delivering the employer brand within the work environment. An HR manager elaborated:

During COVID, we immediately transitioned to remote work and embraced a high degree of flexibility. For individuals with young children, it was essential for them to occasionally take breaks to attend to their kids, and we promptly communicated that this was completely acceptable. It turned out that this gesture was highly appreciated, and everyone navigated through the situation successfully. Ultimately, this employer branding approach had no negative impact on the end–to–end performance of the company and the work conducted remained unaffected. Additionally, offering flexibility during that period also significantly enhanced their sense of autonomy, satisfaction and strengthened their connection with the company. (male, small organization, information technology)

A vast majority of the respondents showcased creative flexibility by reimagining traditional team-building strategies to suit the digital landscape. The shift from off-site activities to virtual alternatives underscored a commitment to nurturing team cohesion and the internal employer brand, even within the confines of remote work environments. HR managers actively engaged in devising strategies and creative activities, such as virtual quizzes and the implementation of a dedicated week of happiness. Moreover, addressing the inherent isolation of remote work, HR managers took proactive measures to cultivate social connections among team members. Introducing initiatives like digital coffee breaks and a secret friend activity, these efforts underscored the crucial role of human connection in maintaining a positive work environment. Important to note is that HR managers consciously aligned these virtual activities with the organization's employer brand, ensuring that these initiatives authentically reflected its values and identity. This strategic alignment sought to reinforce a collective sense of belonging and purpose among employees, bridging the gap created by the absence of traditional workplace employer brand interactions. One HR manager reported:

That was not always possible, but when we organize initiatives for our staff, we always make it a point to say: we are a caring organization for that reason. The year–end gift is to be understood in that context because you have shown a great deal of dedication, and we also want to extend warmth to you, just as you show warmth to our patients. So, we always try to link it to our employer brand framework, in terms of values. (male, large organization, social sciences)

*Subtheme 1.b.3*: *More focus on the internal vs external employer brand*. Moreover, during the pandemic, some participants stated that the internal aspect of employer branding became more apparent. Two HR managers clarified that amidst the unprecedented challenges posed by the global health crisis, the significance of fostering a strong internal employer brand gained heightened recognition.

During the pandemic, we were more focused on our internal brand and ourselves. Before COVID–19, we were focusing on: "How are we perceived in the labor market?" But now, we pay attention to: "How do our employees look at us? How are our employees doing?" So, we are more focused on creating a connection with our employees. (male, large organization, information technology)

We need to focus more on retention instead of recruiting through external employer branding. We have a good external employer branding strategy, but we need to work on internal employer branding, which became clear during the pandemic. (female, large organization, medicine)

*Subtheme 1.b.4*: *Common enemy feeling.* A minority of the participants mentioned the feeling of a common enemy among their workforce, which encouraged employees to connect with the employer brand and enhanced cohesiveness.

## Main theme 2: Internal communication

**Theme 2a: COVID-19 imposes challenges.**   *Subtheme 2.a.1*: *Impossible to implement traditional employer brand communication approach.* Based on our analysis, we can conclude that HR managers during a crisis still communicated about the employer brand. However, nearly all the HR managers (31) expressed that communication about the internal employer brand was different than during stable times. One HR manager summarized: "*During the pandemic, our employer brand strategy was limited to the most necessary subjects. There was zero interaction and interpersonal contact, which made it difficult to deliver the employer brand to employees.*" (female, small organization, textile).

Moreover, organizations delivering the employer brand mainly through physical presence reported being highly challenged. Physical presence and activities to radiate the internal employer brand were restricted in organizations where employees had to work remotely. For nearly all organizations communication about the internal employer brand was now mainly delivered through digital channels, such as emails, digital newsletters, and intranet (besides some manufacturing organizations where employees had to come to the workplace). Some HR managers put forward that fun activities, interpersonal conversations, and real-life contact were crucial actions for them to deliver the internal employer brand, but this was not possible because of the context. An HR manager reported:

Transmitting the employer brand physically has completely disappeared nowadays. For me, two days a week have become available for me where I used to have face–to–face interactions with the employees about the employer brand. Moreover, conveying the values and work practices digitally is quite challenging, especially ensuring that it resonates effectively with individuals. That is why we have dedicated a significant amount of time to fine–tune those conversations digitally. Even though they were conducted over the phone or virtually, we invested a lot of effort to give it extra attention. (female, small organization, finance)

*Subtheme 2.a.2*: *Less bottom-up feedback.* As a result of the lack of interpersonal communication, HR managers expressed difficulties in not knowing how employees receive bottom-up communication regarding the employer brand. Moreover, some HR managers even thought that this could develop misinterpretations of the internal employer brand. One participant summarized:

Communication was always a challenge. This has become even more difficult due to the COVID–19 pandemic. Nowadays, a message is mainly communicated digitally. Therefore, it is difficult for us to know if the employer brand message is understood and interpreted correctly. (male, small organization, information technology).

*Subtheme 2.a.3*: *Navigating communication overload for internal employer branding.* Some participants also pointed out that employees may be overflowed with internal communication. HR managers suggested that they communicated frequently through emails, online meetings, and intranet updates, and at a faster pace. The communication is not solely centered around internal employer branding, however, four HR managers indicated that employees had difficulties in differentiating what is important to them. They suggested that employees may have a

lack of knowledge of the internal employer brand as they cannot differentiate relevant internal employer brand communication from other communication.

**Subtheme 2b: COVID-19 creates opportunities.** *Subtheme 2.b.1*: *Emphasis on warmth and care in employer brand communication*. COVID-19 also created opportunities for communication about the internal employer brand. First, some participants (11) recognized that the content of the communication has developed towards a softer side. Internal employer branding communication stemming from the organization was directed more towards improving employees' trust in the internal employer brand rather than controlling their employer brand behavior. Therefore, communication to employees regarding the internal employer brand appeared to be geared towards emanating warmth and care. This was achieved by emphasizing initiatives aimed at encouraging employees to take breaks, disconnect, and prioritize their mental health. HR managers mentioned focusing on coming across as friendly and attentive when they communicated about the employer brand. An HR manager stated that: "*The channels remain the same, but the language has changed. There has been much more communication around well-being.*" (female, large organization, information technology) Moreover, in another organization an HR manager reported how they communicated in a warm style:

> I found it a very powerful message when your CEO says, "At five o'clock, you can close your laptop." So, people continued to give, and that is also the pitfall of such an organization where everyone goes for their job at 120%. People can lose themselves. And as an organization, we have a very important role in protecting them. So now the employer brand message is: "Dear people, take your Christmas vacation. Disconnect, take a break, because next year will also be busy, and we do not yet know what awaits us." (female, large organization, transportation, distribution, and logistics)

*Subtheme 2.b.2*: *Implementation of new digital communication strategies*. The adoption of these new digital communication strategies not only helped HR managers overcome challenges but also created opportunities to strengthen the internal employer brand. First, HR managers reported innovative ways of delivering the internal employer brand by implementing digital activities, such as e-aperitives, (online) quizzes, etc. One participant explained that these actions facilitated the transmission of the employer brand:

> The distance or the absence of the employees in the office was a real challenge. We tried to respond to this, so we invested in tools (e.g. info sessions, fun online activities, etc.), tips and tricks, to be able to maintain the delivery of the internal employer brand. (male, large organization, science, technology, engineering, and mathematics)

Furthermore, two respondents mentioned the transition to videos showcasing the employer brand. As one respondent elaborated:

> We actually switched to video messages at that time, which a colleague and I recorded every Monday. It was a fifteen to thirty–minute session where we provided an update on what was happening in the organization and decisions made regarding instances about the employment experience and employer brand, and, especially in the beginning, it was very relevant. (male, large organization, information technology)

Additionally, to counteract the overflow of communication, two HR managers mentioned installing "target group communication" management. Hereby, they target certain groups of

employees and communicate only what is relevant to them to reduce confusion and to support employees in discerning internal employer branding communication.

According to HR managers from various companies, the emphasis on two-way communication has played a crucial role in cultivating a positive and engaged work environment amid the pandemic. HR managers have implemented daily Zoom meetings as a means for employees to express themselves and share their feelings, contributing to a more open and supportive atmosphere. This sentiment was echoed by another respondent who highlighted that the structured yet frequent nature of these meetings ensures the prompt dissemination of information about the employer brand. In addition, some HR managers have introduced employee surveys to serve as a platform for gathering input and feedback about the employer brand and to minimize the risk of misinterpretations about the internal employer brand. Respondents have underscored an enhanced sense of employee involvement, emphasizing the significance of openness and the creation of opportunities for colleagues to connect and share their experiences about the organization, employer brand, and work difficulties.

Furthermore, respondents mentioned the importance of timely and transparent communication about the organization and employer brand during the pandemic. Respondents across various organizations underscored the significance of these strategies. They stated to foster transparency by openly communicating about personnel changes, successes, but also failures and holding Q&A sessions. Furthermore, a notable aspect emphasized by the respondents is the introduction of CEO-led info sessions as part of HR-driven initiatives. These sessions, led by top leadership, served as a cornerstone of transparent communication. HR managers mentioned that they collaborated with CEOs to share insights openly, providing employees with valuable perspectives on the organization's strategies, challenges, and successes. A respondent elaborated: "*We also have free talks, we, as CEOs, spend a specific time in a room, and people can ask us any question, for instance regarding the work practices, etc., to which we provide answers.*" (male, large organization, science, technology, engineering, and mathematics)

Respondents further highlighted the proactive measures taken to ensure accessibility. Beyond conventional communication channels, HR managers mentioned a low-threshold approach, making direct contact details of management readily available to employees, as well as organizing meetings between new employees and the CEO. An HR manager stated:

> We are going to set up something new, an idea that is emerging, is a breakfast with the CEO and the owner, where they can learn about the employer brand. We will invite all newcomers who have started in the past two months. They will then attend a breakfast session from 8–10 persons, where they can ask questions to the CEO or the owner. The aim is to continually create that bond, emphasizing that hierarchical lines should remain very limited, very small. (male, large organization, science, technology, engineering, and mathematics)

### Theme 3: Leadership

**Subtheme 3a: COVID-19 imposes challenges.** *Subtheme 3.a.1*: *Loss of employee monitoring*. One-third of the respondents reported that some supervisors experienced difficulties monitoring employees and verifying if they were behaving according to the employer brand. This challenge was particularly evident in scenarios where remote work became prevalent, leading to concerns about maintaining a consistent alignment between employees' actions and the organization's articulated employer brand values.

**Subtheme 3b: Covid-19 creates opportunities.** *Subtheme 3.b.1*: *Key role in transferring and radiating the employer brand*. HR managers (17) underscored the critical role of supervisors in the transfer and dissemination of the employer brand, emphasizing that it extends

beyond conventional communication of employer brand principles. Instead, HR managers recognized that supervisors play an indispensable role in embodying these principles through their day-to-day actions and interactions with their teams. This perspective aligned with the broader acknowledgment of the significance of supervisors as role models within the organizational structure. By enhancing their ability to align their actions with organizational values, supervisors became influential figures, according to HR managers, who contributed significantly to the establishment and perpetuation of the desired employer brand culture.

Additionally, HR managers highlighted the adaptive measures taken by supervisors in adjusting communication channels. This proactive adjustment involved increased engagement in conveying crucial information to their teams, such as employer branding communication. The emphasis is placed on the positive opportunities that arise from enhanced online interaction within teams. This shift not only facilitated effective communication, but also opens avenues for teambuilding, collaboration, and improved overall teamwork.

*Subtheme 3.b.2*: *Assignment of additional tasks*. To deal with these challenges during the pandemic, many HR managers mentioned assigning additional tasks to the supervisors. First, as stated above, HR managers had to alter their approach toward new employees as they had more difficulties connecting with the employer brand. Two HR managers explained that they instructed supervisors to give special attention to the new employees and deliver the internal employer brand (in a virtual setting), by clearly demonstrating the employer brand when interacting with (new) employees.

> We instructed supervisors to talk and ask (new) employees about the employer brand. And that does establish and fosters that connection with the employer brand. I'm a manager myself and it certainly hasn't gotten any easier in the pandemic. It requires more organization, more structure, more scheduling, and dedicating extra attention to do this. (female, large organization, Sector: Agriculture, Food, and Natural Resources)

Additionally, some HR managers reported that they instructed supervisors to reduce possible confusion about the internal employer brand and prevent the development of multiple interpretations. To this end, supervisors were asked to have more personal conversations with their employees and to behave in line with the internal employer brand.

Furthermore, HR managers mentioned that they encouraged employees to ask for more clarifications about the organizational culture and employer brand to improve their understanding of the employer brand. It seems that during the pandemic, 16 HR managers allocated supervisors with the task of delivering the employer brand to their subordinates. They instructed supervisors to have enough personal (digital) conversations to clarify and familiarize employees with the internal employer brand.

Summarized, HR managers acknowledged that they gave supervisors more responsibilities regarding the internal employer brand and expected certain behaviors, such as more contact with new employees, dissolving misinterpretations regarding the internal employer brand, and attending information sessions. By delegating the employer brand tasks to the supervisor, HR managers could focus more on the strategic aspect of internal employer branding, for example, building and creating the internal employer brand.

*Subtheme 3.b.3*: *Switch to coaching and supporting*. Respondents emphasized the pivotal role of the touchpoint supervisors, in showcasing a more people-centered leadership style, especially during the pandemic. HR managers advised supervisors to transition towards a more understanding and supportive leadership style when delivering the employer brand and to trust employees to implement the employer brand accurately. During the pandemic, supervisors were directed to emphasize people skills in delivering the employer brand, as highlighted

by the 12 HR managers. This shift was reflected in their active attention to team members, going beyond traditional methods. Notable initiatives included personalized virtual team-building sessions, like online coffee meetups and fitness challenges, aimed at conveying organizational values and fostering a cohesive internal environment.

HR managers highlighted the importance of supervisors understanding employee needs, fostering a culture of mentorship, and playing a crucial role in providing support during challenging times. This involvement was seen as integral to reinforcing the organization's commitment to employee welfare and the employer brand. Recognizing the significance of supervisors prioritizing their teams, HR managers highlighted its positive impact on conveying an image aligned with employees' values.

The participants noted that supervisors, in response, organized enjoyable initiatives to enhance the internal delivery of the employer brand. This proactive approach, aligned with a people-centered leadership style, demonstrated a commitment to workforce growth and cohesion. Supervisors actively contributed to shaping the internal employer brand by providing support and fostering a skilled and closely-knit team. Despite these positive shifts, HR managers reported that the adoption of a different mode of delivering the employer brand was not uniformly evident for every supervisor. An HR manager mentioned:

> Some supervisors had a more controlling approach when supervising and supporting employees on radiating the employer brand. During the pandemic, supervisors were forced to adapt their behavior–as employees were working from home and direct supervision was not possible. This was a challenge for supervisors–as they were forced to develop people skills and transform their leadership style. (male, large organization, hospitality)

*Subtheme 3.b.4*: *Installments of training sessions*. To alleviate these assigned responsibilities regarding the internal employer brand, respondents disclosed installing additional training sessions. Almost 13 of the respondents installed training sessions in delivering the employer brand. One HR manager elaborated on the content of the info session, as follows: "*During this pandemic, we want supervisors to be the example of our employer brand. Therefore, we installed training in the context of: 'How do you carry those employer brand values during these difficult times? This was not easy to install during the pandemic.*" (female, large organization, agriculture, food, and natural resources).

**Exploring the context: Warm vs competent employer brands.** To incorporate contextual elements, we investigated whether distinct types of employer brands—warm versus competent, encountered different challenges and opportunities. We initiated a contextual analysis by first soliciting descriptions of internal employer branding and the content of employer brands during the interviews. Organizations that articulated a familial, warm, or social emphasis in their descriptions were categorized as warm employer brands. Conversely, those highlighting aspects such as training opportunities, employee growth, and competence development were classified as competent employer brands. Out of the 37 organizations analyzed, we successfully categorized 28, identifying 11 as warm employer brands and 17 as competent employer brands. This categorization formed the foundation for further exploration, allowing us to investigate whether and how these distinct employer brand types encountered unique challenges or opportunities amidst the pandemic.

*Content*. Examining the content of these two types of the employer brand, it is noteworthy that challenges across the two types of employer brands appear largely similar. However, the key difference lies in their content. Competence-oriented employer brands prioritize formality, competencies, and efficiency within a professional framework. They focus on efficiency, innovation, and digital transition. In contrast, warmth-oriented brands emphasize personal

connections, care, well-being, employee recognition, and a positive atmosphere during the pandemic. They present well-being as part of a broader positive branding strategy, utilizing symbolic language and digital communication. The communication style is informal, fostering an open atmosphere and focusing on individual well-being to enhance team culture and relationships. Despite facing common challenges, each approach customizes its employer brand transfer and communication style according to its unique employer value proposition, reinforcing its distinct employer brand identity.

*Approach*. Both competence-oriented and warmth-oriented employer brands showed a strong commitment to internal branding during the pandemic, emphasizing on engagement and employer brand values. Competency-focused organizations strategically addressed the challenges, prioritizing innovation, and preserving organizational culture. In contrast, warmth-focused organizations prioritized human contact, informal interactions, and genuine connections for internal branding. In terms of internal communication, both recognized the impact of digital transformation, demonstrating flexibility, and valuing a balance between formal and informal communication. Technological acceptance was crucial for both, with the competence-oriented approach prioritizing innovation and transparency, while the warmth-oriented approach highlighted spontaneous interaction, positive feedback, personal connections, and emotional engagement. Regarding leadership, both valued regular communication from supervisors, support for employees, and bottom-up initiatives, but competence-focused organizations emphasized adapting to changing work environments by providing training sessions, while warmth-focused organizations stressed leadership in fostering connections and supporting employees through pandemic-induced changes. In conclusion, while each employer brand faces similar challenges, they tailored their approach to align with their unique employer value proposition, thereby reinforcing a distinctive employer brand identity within the organization.

**Member check.**   To validate our findings, we conducted a member check with HR managers from various organizations. All our results were affirmed and supported by the participants in the member check (refer to 'S5 Table'). The findings highlight the diverse experiences and strategies among HR managers, showcasing adaptability to unique circumstances. HR managers from the member check confirmed challenges in aligning employees with the employer brand and onboarding new hires to acquaint them with the employer brand. Participants acknowledged the need to strengthen their employer brand, with the pandemic serving as a catalyst for increased focus on internal employer branding. HR managers recognized the importance of reinforcing retention strategies during the pandemic. While not universally endorsed in the interviews, the "common enemy" sentiment also garnered limited support in the member check. Traditional strategies became less applicable in virtual settings, prompting the implementation and endorsement of new communication strategies, such as warmth-centered content and "target group communication management." HR managers in the member check emphasized the pivotal role of supervisors in conveying the employer brand.

**Employee check.**   To gain insights from employees, we conducted a check involving individuals from the same organizations as those interviewed. Overall, employees validated the findings pertaining to internal employer branding and the touchpoints: internal communication and leadership (see 'S5 Table'). Participants highlighted challenges in perceiving the employer brand due to physical distance, resulting in diminished connection and commitment. Traditional methods of communicating the employer brand to current employees encountered pandemic-related obstacles, as indicated by HR managers and corroborated by employees. The employee check additionally observed a shift toward warmth and care in employer brand communication.

HR managers noted a decline in bottom-up communication, potentially leading to misinterpretations, a sentiment supported by the employee check. Furthermore, HR managers discussed difficulties in monitoring employees and encouraged supervisors to regularly check in with their teams, a point substantiated by the employee check. Employees reported an increase in their interactions with supervisors.

In addressing the need for a different leadership style during the pandemic, HR managers emphasized coaching and support in conveying the internal employer brand. However, many employees did not perceive this as supervisors delivering the employer brand more frequently; instead, they acknowledged increased support through regular check-ins, assistance, and flexibility. This suggests that while positively influencing the employee experience, supervisors' actions may not be explicitly seen as a direct communication or embodiment of the employer brand, but rather perceived as additional check-in time. Employees were cognizant of training sessions for supervisors.

## Discussion

This study aimed to acquire a better understanding of how HR managers perceived and experienced internal employer branding and the touchpoints: internal communication and leadership during a disruptive event. Based on interviews and a member check, we addressed (1) what challenges HR managers experienced during the pandemic regarding internal employer branding and internal employer branding touchpoints; (2) what opportunities HR managers experienced during the pandemic regarding internal employer branding and internal employer branding touchpoints. The employee check provides a first insight into whether and how employees experienced the adopted strategies.

First, this research suggests that a disruptive event, such as the pandemic, imposed challenges regarding internal employer branding, however, despite these challenges, HR managers also adopted different strategies and opportunities. For instance, some organizations paid more attention to establishing and creating a strong internal employer brand. This is remarkable, as according to Anderson et al., [75], we can assume that organizations will prioritize cost management and safety over other HR practices, such as employer branding. In addition, the pandemic was an inflection point for some organizations to rethink their employer branding strategy. Some actively considered how to avoid making "a bad situation worse" and how to alleviate some of the negative influences this crisis had on employer branding [76]. Thus, this study indicated that (internal) employer branding was not disregarded during the crisis, moreover, intentional attention was directed to it.

Second, this study provided some valuable observations related to the touchpoint internal communication about the employer brand. During a crisis, transparent internal communication may play a crucial role in delivering the employer brand among employees as it may foster engagement toward the brand and build a stronger relationship between employees and the organization [77,78]. However, noise can emerge, which may hamper the understanding of the internal employer brand [79]. This is in line with the HR managers' perceptions, as they mentioned that the message about the employer brand sent by the organizations was sometimes interpreted wrongly due to the virtual setting that COVID-19 imposed. According to Kniffin et al., [80], virtual communication develops the risk of misunderstandings or inconsistencies. These misinterpretations, the possibility of multiple internal employer brands, or inconsistency between the desired and the perceived employer brand may have counterproductive responses, such as lower well-being, lower commitment, and higher turnover intention [81]. Moreover, organizations reported a decline in bottom-up feedback about the employer brand. Therefore, HR managers received limited information about how to improve their

employer brand to remain attractive to employees [6]. To minimize this, some HR managers provided clear and short communication and installed online feedback surveys, which could assist in receiving bottom-up information.

Additionally, this study has shown the valuable role of supervisors in developing, transferring, clarifying, and "living" the employer brand during a crisis. Supervisors in times of COVID-19 may have seen an expansion of their responsibilities regarding employer branding. This is in line with He et al., [82] suggesting that adequate leadership and supervisor support can boost employees' motivation and increase employee engagement during turbulent times [55]. Supervisors may have become more responsive to help employees adjust to the employer brand, new situations, and work environments. For instance, supervisors were ascribed with the responsibility to dissolve misinterpretations about the employer brand among employees. Gilani and Cunningham [6] already suggested the importance of leadership in the employer branding literature, however, it seems that in turbulent times the value of leadership has become more apparent for organizations. Supervisors, not solely during a crisis, might have a key and operational role in delivering the internal employer brand in the organizations. As such, HR managers may have more time to focus on the strategic aspect of internal employer branding.

Finally, our results showed that strategies and opportunities adopted to navigate the challenges during the pandemic varied based on their employer brand orientation, specifically warm or competent [12,83]. Both orientations encountered difficulties related to new strategies, onboarding, remote work, and maintaining a connection with the employer brand. Despite these shared challenges, competence-focused organizations strategically approached these issues, emphasizing innovation and the preservation of organizational culture. In contrast, warmth-focused organizations prioritized human contact and informal interactions, emphasizing the importance of genuine connections for internal employer branding. This observation may suggest, though it requires further investigation, that the employer brand plays a pivotal role in shaping distinct priorities and approaches when addressing internal employer branding in times of crisis. Moreover, this study indicates organizations' commitment to living the employer brand, staying true to it, and avoiding radical changes during a disruptive event.

## Theoretical implications

Most of the existing employer branding literature focuses on external employer branding and neglected to pay attention to internal employer branding. However, literature highlighted the valuable role of internal employer branding in employees' attitudes, such as more employee engagement and lower turnover attention [9,30,31,84], which can be related to firm performance. This research fills a literature gap by displaying how specific aspects of internal employer branding and the touchpoints internal communication and leadership were experienced during unstable times. Moreover, the study is in response to the call of Lievens and Slaughter [12] that insights are required on how a disruptive event can transform an employer brand, and it enriches the internal employer branding literature.

Thirdly, this study underscores the critical role of HR managers as central figures in navigating internal employer branding during crises, with a specific focus on the COVID-19 pandemic [11]. Theoretical implications of these findings provide insights into the decision-making processes of HR managers concerning internal employer branding during crises. The research underscores the strategic engagement of HR managers, their innovative solutions, and their efforts to uphold the employer brand in challenging times.

Additionally, this research extends the internal employer branding literature by exposing that during a crisis HR managers reported that they still focused on building an effective

internal employer branding strategy, next to the procurement of the external employer branding strategy. Although existing literature studied mainly external employer branding, our research has displayed that constructing an effective internal employer brand is also important for organizations, even in turbulent periods. As such, we highlighted the discrepancy between what organizations are interested in (internal and external employer branding) and what has already been explored (mainly external employer branding) in literature.

Furthermore, the importance of leadership for employer branding has become apparent in this study. Existing literature, such as Gilani and Cunningham [6], has displayed the importance of implementing employer branding in the entire organization. In this context, our results indicated the crucial role of supervisors. This study confirms the current tendency to devolve operational HRM tasks to supervisors [55] as our results suggest that supervisors can fulfill several tasks concerning internal employer branding.

Finally, according to the interviews conducted for this research, organizations with various types of employer brands (such as warm vs. competent) encounter similar challenges during a crisis but may opt for distinct strategies. The employer brand or employer value proposition serves as a crucial boundary condition during the pandemic, shaping the organization's response to external uncertainties and internal dynamics.

## Practical implications

In sum, we believe that the initial qualitative findings of this study should be strengthened by insightful future research before conclusive practical recommendations can be made. However, we summarize the preliminary findings and carefully convert them into the following suggestions for practitioners.

First, organizations should be mindful of the influence a crisis can have on their internal employer branding and the mechanisms used to convey the employer brand. The study suggests that crises, like the COVID-19 pandemic, pose challenges but also offer opportunities for enhancing internal employer branding. It is essential for organizations to be cognizant of these influences and assess their implications.

Second, according to literature, internal employer branding can be a critical organizational resource for retaining employees in a labor market. Therefore, organizations are advised to persist in disseminating their internal employer brand throughout challenging times. Additionally, crises can prompt organizations to reflect on and potentially reconsider their overall approach and policies related to employer branding.

In addition, organizations may need to reassess how they communicate the internal employer brand through internal channels and supervisors during turbulent times. Strategies that were effective in stable periods might prove ineffective, necessitating a contemplation of their continued relevance. HR managers should evaluate the applicability of existing strategies and consider alternative approaches. For instance, maintaining a warm and friendly tone in employer brand communication can be valuable during times of high uncertainty in the workplace [82]. Furthermore, the use of new digital communication strategies may help ensure a clear and consistent employer brand message during disruptive events.

Finally, recognizing the significance of supervisors during unstable times, organizations may consider assigning additional responsibilities to supervisors, including the task of explaining the employer brand to new employees and addressing misinterpretations. This allows HR managers to focus more on the strategic aspects of internal employer branding. Providing support to supervisors in developing interpersonal skills becomes crucial during turbulent periods, emphasizing the need for a shift towards interpersonal skills over technical skills when conveying the employer brand. Although HR managers instructing supervisors to convey the

employer brand, findings from the employee check revealed that participants did not necessarily perceive supervisors as embodying the employer brand. However, it is noteworthy that while supervisors might not have been seen as direct embodiments of the employer brand, participants expressed appreciation for their consistent check-ins, flexibility, and support. This suggests that, even if not explicitly recognized for employer brand representation, supervisors played a crucial role in fostering positive employee experiences through their supportive actions.

## Limitations and future research

Although this research provides, insights into internal employer branding during COVID-19, it has some limitations. First, this research is based on interviews and a member check with a select group of HR managers, which consequently indicates that the results do not apply to all organizations. Moreover, this study only looked into Belgian organizations; these insights might vary between different countries because different regulations by governments were installed and a different context could be in place. Hence, future research on internal employer branding in times of crisis with cross-national samples is needed to create an overview of how COVID-19 affected organizations all over the world.

Second, our qualitative findings cannot establish causality. Therefore, our research cannot exclude other aspects that possibly influence internal employer branding. Moreover, our study cannot conclude with certainty that these challenges and opportunities are caused by COVID-19. As such, quantitative analysis is required to unravel this further. Furthermore, existing literature has not extensively discussed how supervisors can influence the internal employer brand and what leadership skills are required for transmitting the employer brand among employees. However, these results suggest an important role for the supervisor, therefore, research should be carried out to gain insight into the role of supervisors in employer branding.

Furthermore, this study reveals that different types of employer brands, specifically those characterized as warm versus competent, encounter similar challenges amid crises but may adopt unique strategies based on their employer value proposition. Different theoretical frameworks, such as Instrumental-Symbolic Framework [27], could be used to incorporate a deeper understanding of how crisis events impact diverse employer brand orientations. Based on initial interviews and considering past literature, the decision was made to prioritize internal communication and leadership as touchpoints. However, we may have overlooked other important actions for internal employer strategies during times of crisis. As such, future research could also delve into (other) mechanisms or touchpoints through which organizations strategically align their internal branding efforts in response to crises, shedding light on the dynamic interplay between external uncertainties and internal organizational dynamics. Unpacking the influence of these elements on organizational responses to crises remains an essential area for exploration. Future studies could investigate how variations in employer brand messaging and value propositions shape decision-making processes, internal communication strategies, and overall resilience during challenging times.

Despite HR managers and the literature emphasizing the importance of supervisors in transferring the employer brand to employees, the participants in the employee check did not particularly perceive this. They rather perceived this as additional check-in time with their supervisor. This could suggest the need for caution when depending on employees' perspectives regarding employer branding efforts, as additional factors like leadership styles may play a role in how employees assess internal employer branding initiatives. Moreover, this incongruence between HR expectations and employee perceptions warrants further investigation.

Future investigation may be required to explore the precise factors underlying this discrepancy, which could potentially stem from leadership style or the dynamics of supervisor-employee interactions. Investigating how various leadership styles, such as authoritarian, democratic, or coaching leadership, influence employees' interpretations of internal branding efforts is essential. For example, under a coaching supervisor, frequent check-ins can reinforce the perception of internal employer branding efforts by fostering supportive and collaborative interactions especially during challenging periods. Conversely, under an authoritarian supervisor, the intensified control and directive nature of conversations may play a role in how employees perceive the internal employer brand initiatives. Understanding these dynamics can provide valuable insights into optimizing the role of supervisors in internal employer branding and bridging the gap between managerial expectations and employee experiences.

Although, we conducted an employee check to explore the perspective of the employees, we mainly investigated the perspective of the HR manager. However, due to the virtual setting of most participating organizations and the physical distance between HR managers and employees, the perceptions of HR managers could potentially be clouded or the impact on employees could be perceived differently by the respondents. Nishii et al., [85] highlighted the potential discrepancy between HR managers' actual practices and employees' perceptions of these actions. Therefore, future research can take on a multilevel approach and investigate the perspective of HR managers, employees, and supervisors and compare these perceptions. This approach would provide a more nuanced and holistic view of internal employer branding practices, allowing for a deeper exploration of the dynamics between organizational strategies and their perceived impact on different levels within the workforce.

## Supporting information

**S1 Table. Demographic information on the cases and respondents of the interviews, member check, and employee check.**
(DOCX)

**S2 Table. Minimal dataset including data availability statement, ethical elements, and data-analysis process.**
(DOCX)

**S3 Table. Overview and description of the (preset) themes.**
(DOCX)

**S4 Table. Relevant excerpts from the interviews per theme.**
(DOCX)

**S5 Table. Overview of research findings, member check, and employee check.**
(DOCX)

## Acknowledgments

We would like to thank Kerlijn Dangreau, Florence Willaert, and Kenny Claes for their help in conducting and transcribing the interviews.

## Author Contributions

**Conceptualization:** Marthe Rys, Eveline Schollaert.

**Data curation:** Marthe Rys.

**Formal analysis:** Marthe Rys.

**Investigation:** Marthe Rys.

**Methodology:** Marthe Rys.

**Project administration:** Marthe Rys.

**Resources:** Marthe Rys.

**Software:** Marthe Rys.

**Supervision:** Eveline Schollaert, Greet Van Hoye.

**Validation:** Marthe Rys.

**Visualization:** Marthe Rys.

**Writing – original draft:** Marthe Rys.

**Writing – review & editing:** Marthe Rys, Eveline Schollaert, Greet Van Hoye.

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
