## [Decision Letter · Decision Letter 0]

21 Nov 2023

PONE-D-23-17334Living the employer brand during a crisis? A qualitative study on internal employer branding in times of the Covid-19 pandemic.PLOS ONE

Dear Dr. Rys,

Thank you for submitting your manuscript to PLOS ONE. After careful consideration, we feel that it has merit but does not fully meet PLOS ONE’s publication criteria as it currently stands. Therefore, we invite you to submit a revised version of the manuscript that addresses the points raised during the review process.

We look forward to receiving your revised manuscript.

Kind regards,

Nikolaos Georgantzis, Dr.

Academic Editor

PLOS ONE

Journal Requirements:

Did you know that depositing data in a repository is associated with up to a 25% citation advantage (https://doi.org/10.1371/journal.pone.0230416)? If you’ve not already done so, consider depositing your raw data in a repository to ensure your work is read, appreciated and cited by the largest possible audience. You’ll also earn an Accessible Data icon on your published paper if you deposit your data in any participating repository (https://plos.org/open-science/open-data/#accessible-data).

3. You indicated that ethical approval was not necessary for your study. We understand that the framework for ethical oversight requirements for studies of this type may differ depending on the setting and we would appreciate some further clarification regarding your research. 

Could you please provide further details on why your study is exempt from the need for approval and confirmation from your institutional review board or research ethics committee (e.g., in the form of a letter or email correspondence) that ethics review was not necessary for this study?

Please include a copy of the correspondence as an ""Other"" file.

5. Please ensure that you include a title page within your main document. You should list all authors and all affiliations as per our author instructions and clearly indicate the corresponding author.

Reviewers' comments:

Reviewer's Responses to Questions

**Comments to the Author**

1. Is the manuscript technically sound, and do the data support the conclusions?

Reviewer #1: Yes

Reviewer #2: Partly

2. Has the statistical analysis been performed appropriately and rigorously? 

Reviewer #1: N/A

Reviewer #2: N/A

3. Have the authors made all data underlying the findings in their manuscript fully available?

Reviewer #1: No

Reviewer #2: Yes

4. Is the manuscript presented in an intelligible fashion and written in standard English?

Reviewer #1: Yes

Reviewer #2: Yes

5. Review Comments to the Author

Reviewer #1: The study topic is interesting and deserves to have a chance as a publication to understand more about employer branding. However, with the current writing, I feel that theoretical background has not been deeply analyzed. I suggest the authors to improve the section as well as the introduction part. The technical details should be expanded and clarified to ensure that readers understand exactly what the researchers studied.

In the materials and methods section, I did not see the interview guides. I am particularly curious on how the "questions were continuously adapted..." (lines 234-235).

Also, not particularly convinced on how the authors doing the population and sample choices. Although it is briefly mentioned in the lines 217 - 220, but when we talked specifically about COVID 19, a lot of business will experience significantly different with others (e.g., hospital workers were busier while many restaurants closed their business; or business in urban, big cities vs in more rural locations).

With a relatively big number of respondents, I would expect more thoroughly results. I am also concerned about the quotes that are proportionally taken from member check, not the respondents. Member check is about confirming the respondents' answers as well as non-response bias.

In the analysis process, as you mentioned that you used Microsoft Word, how to check inter-rater reliability - a measure of the degree of agreement between multiple researchers analyzing the same data? If we are using a qualitative analysis software such as NVivo, the inter-rater reliability can be measured using the Kappa statistic or percent agreement.

Lastly, please be consistent with the citation style and do another round of editing.

Reviewer #2: Dear authors,

I read with great interest your article on internal employer branding during a crisis. I do see the added value of having more in-depth insights in the art and impact of internal employer branding and also during challenging /crisis times. Overall the manuscript reads well, and is logically structured. I appreciate the transparency on the interviewees’ profile in Table 2, and the extent to which the extra 6 interviewees (member check) agreed or not with the results you provided in Table 1. I struggle the most with a lack of clear added value of your study beyond the mere fact that such research is absent, the lack of the viewpoint of internal employees especially given the aim and practical implications that were formulated (now purely based on responses of HR managers), and the lack of more in-depth rich dynamic insights derived from the data (e.g. also distinction between inductively and deductively deduced themes/codes). I will explain these issues hereafter in more detail.

oAdded value of your research and unclarity in research focus

In introduction it remains unclear why we would need research on internal branding strategies in crisis beyond the mere fact that there is no such research. A stronger rationale is needed as for why we expect current research findings on internal employer branding during stable times would not hold in case of crisis or more unstable periods. Maybe an in-depth look at changes in preferred work values or needs of employees might help as perspective to frame it why it would or could look different in unstable times for HR managers on how to make work of internal employer branding. On page 6, lines 129-132 idea of togetherness as more important during crisis can be a starting point, but needs more elaboration. Same goes for touchpoint of leadership on page 7 (lines 157 -162), you mention importance of flexibility by leader and you just name adopting appropriate strategies but any in-depth elaboration lacks of which internal employer branding strategies might then be needed in crisis more or less and why in the sense of likely impact on internal employees’ motivation, satisfaction, ...

After introduction and literature it remains unclear what the exact research focus and question is. Is it about which internal branding strategies are adopted by HR managers during crisis or what impact one might have on employees of adopting same or changed internal branding strategies in terms of internal communication/leadership?

If you want to contribute with your research to practice by being able to guide organizations in handling their internal employer branding during future crisis (end of introduction), I do not understand how you can do that validly by only questioning and studying the view point of HR managers. At least we would need to have an idea of the impact of certain adopted internal employer branding strategies by studying then the voice and perspective of employees themselves (what were there needs, changed work anchors or work values, how did they experience the adopted internal employer branding during crisis?) . Also at the end of the literature section on ‘ employer branding’ (line 90-92) you repeat that research on internal employer branding is scarce and that therefore you will focus on exploring internal employer branding during unstable times, but why this is relevant and e.g. hence different from stable times remains untouched.

oClarity on distinction between internal brand and internal employer brand

The authors clearly describe in beginning of manuscript the difference between internal brand and internal employer brand (lines 101-103). However, in literature but especially in the results there are many statements that point toward internal brand in terms of how employees embrace it and translate it in how they interact with customers, how they do their work instead of reflecting what makes the culture, the work practices, the employer in general unique and worth to stay with in a motivated way as employee. Internal brand instead of internal employer brand: For example line 150 in literature. For example in results lines 325-329.

The results with regard to more supportive leadership point at some places to ‘when supervising and supporting employees to radiate the employer brand’, but the question presents itself is this about radiation to external stakeholders (hence internal brand) or just in general a more people-centered leadership style regardless of the link with bringing alive the internal employer brand. More in-depth elaboration on these results with how then precisely this affects managing the internal employer brand could remove these doubts. In the next point some more examples will be shared where the richness of the data is not translated in the results.

oLack of in-depth insights, more richness needed

-In subtheme 2b the result is presented that the internal employer brand is communicated more in a warm way (line 416). It is nice that the authors also name a case in which misinterpretations on internal employer brand could be solved with short wording. Then an in-depth analysis could reveal if this short style is to reconcile with the warmth? Or is this not an appropriate question as both strategies may relate to different types of internal employer brands. It would be more nuanced to allow the reader to have some basic idea of the core elements of the internal employer brands that were studies along the 36 respondents. We could argue that internal employer brands being higher on warmth and togetherness also before crisis might have experienced different challenges and opportunities versus internal employer brands being focused on e.g. efficiency or competency or … This richness as well as the dynamic perspective can be much more developed and is needed to more validly understand any contextually conditions that might have affected the results for example with regard to focusing more on togetherness, on warmth, ... Based on the definition of internal employer brand one could wonder if there are any instrumental or other symbolic factors that came to the surface in terms of posing more challenges or opportunities.

-It would enhance the internal validity if themes/codes could be separated that were deductively versus inductively derived, e.g. could be added to Table 1. Now the results read as if only the lens of prior literature on internal communication and leadership as essential touchpoints were followed to analyze the transcriptions with no openness toward new findings, like other touchpoints that might have become more essential just because of the crisis (e.g. any instrumental attributes) beyond communication or leadership, or same touchpoint but managed differently e.g. like you bring in results in terms of bringing more warmth in communication and leadership (lines 416 and line 430, 433).

oView of employees needed

One of the results is that HR managers identified that employees had difficulties in recognizing the employer brand (line 293/294). But the question presents itself if this is problematic during crisis and why so. So this result could gain much more relevance if it could be linked to the impact on the employees and why it matters that employees had difficulties in recognizing the employer brand. Were they more easily seduced to leave or did it affect the culture in a negative way or ….

Now the results read as what HR managers did and struggled with in terms of internal employer branding efforts, and that they attached importance to internal employer brand even in crisis. While the latter is a useful finding, the question that needs to be equally solved is then ‘and does it matter in times of crisis to focus or re-focus on internal employer brand and to do it differently with e.g. more warmth like you describe. In the limitations section it is acknowledged that the adoption of certain internal employer branding strategies and its evaluation might differ in the perspective of HR managers and employees, but simply adding this as a limitation does not overcome the impossibility to make claims on guiding organizations on internal employer branding during crisis if you have not tested how the adoption was perceived in the first place by employees (was effort noticed) and secondly how was it evaluated (was effort beneficial for certain outcomes like retention) or were there other needs of employees that were or were not addressed by the internal employer branding efforts during crisis period which might reveal other meaningful internal employer brand challenges during crisis? Therefore, additional data collection is really needed to embrace the experience of employees on the efforts done by HR managers. The latter, the sole focus of the study now, risks missing valuable insights on how the internal employer brand efforts pays off, and is too descriptive in nature like it is brought now in this version of the paper. In-depth rich contextual sensitive and dynamic perspectives can add more valid contributions to the internal employer branding literature than what is currently available in the paper.

-Minor issues

In Table 2 IN-8 is labeled with age of 14, that seems a typo as in line 213 minimum age is 25.

Some spacing issues in sentences resulting in words that stick together like on lines 62 guideorganizations, or line 75 fitstheir or line 569 ondisseminateing

I wish the authors good luck with their further work on internal employer branding and look forward to see more rich and nuanced results.

6. PLOS authors have the option to publish the peer review history of their article (what does this mean?). If published, this will include your full peer review and any attached files.

Reviewer #1: No

Reviewer #2: **Yes: **Diane Arijs

---

## [Author Response · Author response to Decision Letter 0]

15 Jan 2024

1. Thank you for submitting your manuscript to PLOS ONE. After careful consideration, we feel that it has merit but does not fully meet PLOS ONE’s publication criteria as it currently stands. Therefore, we invite you to submit a revised version of the manuscript that addresses the points raised during the review process.

Firstly, we would like to express our gratitude for your honesty and transparency in writing this review. Additionally, we want to thank you for providing us with the opportunity to revise our paper. After thoroughly reviewing the feedback, we decided to resubmit the paper. Your raised points played a crucial role in significantly improving our work.

2. When submitting your revision, we need you to address these additional requirements. Please ensure that your manuscript meets PLOS ONE's style requirements, including those for file naming. The PLOS ONE style templates can be found at https://journals.plos.org/plosone/s/file?id=wjVg/PLOSOne_formatting_sample_main_body.pdf and 

Thank you for providing the additional requirements and the links to the PLOS ONE style templates. I thoroughly reviewed the guidelines and ensured that my manuscript and title page meets the specified style requirements, including proper file naming. I appreciated your guidance, and I have made the necessary revisions accordingly.

Thank you for sharing this valuable information. I highly appreciate the importance of open data practices. However, I'm currently unable to deposit my raw data in a repository as my study participants have not given consent for such sharing. I want to respect their privacy and confidentiality. However, as I do recognize the value of open research practice, I have chosen to deposit a minimal data set in the supplementary materials (see Supporting information ‘S3_Tab’) and displayed some relevant excerpts from the interviews (see Supporting information ‘S5_Tab’). This includes relevant excerpts of the interview transcripts organized by codes. 

4. You indicated that ethical approval was not necessary for your study. We understand that the framework for ethical oversight requirements for studies of this type may differ depending on the setting and we would appreciate some further clarification regarding your research. Could you please provide further details on why your study is exempt from the need for approval and confirmation from your institutional review board or research ethics committee (e.g., in the form of a letter or email correspondence) that ethics review was not necessary for this study? Please include a copy of the correspondence as an ""Other"" file.

Thank you for your insightful comment and your request for additional information regarding the ethical approval of my study. I understand the importance of ethical considerations in research and appreciate the attention given to this matter. Allow me to provide further clarification on why ethical approval was deemed unnecessary for my study. 

First of all, it is crucial to highlight that my research is conducted with a rigorous commitment to ethical principles. I have thoroughly examined the ethical code of my faculty, ensuring that my study adheres to its guidelines. The study is meticulously shaped with a deep understanding of the ethical guidelines established by the faculty. Specifically, the procedures for securing voluntary informed consent from participants, ensuring anonymity during data collection, and allowing participants to withdraw without consequences are intentional reflections of the explicit directives outlined in the faculty's ethical code.

Furthermore, I have included a segment from my correspondence with the institutional review board, confirming their agreement that formal ethical approval was unnecessary for the study. The attached "Other" file contains the official documentation supporting this exemption. The rationale for this decision stems from the qualitative nature of my research, which involved interviews with HR managers regarding employer branding during the pandemic. Given the characteristics of the study, I concluded that it falls into the category of low-risk research. The research design adopts a non-invasive, non-sensitive approach, avoiding loaded questions to mitigate potential harm or discomfort to participants. Participation in the study is entirely voluntary, with participants receiving detailed information about the study and providing informed consent. They understand that they can withdraw at any point without facing negative consequences. Additionally, the data collection process ensures participant anonymity, with all collected data anonymized and securely handled to protect confidentiality. This comprehensive ethical approach aligns with the principles of safeguarding participants' rights and well-being in research, contributing to the overall integrity of the study.

I trust this additional information addresses your concerns. If you have any further questions or require additional documentation or correspondences, I would be happy to provide them. 

5. In your Data Availability statement, you have not specified where the minimal data set underlying the results described in your manuscript can be found. PLOS defines a study's minimal data set as the underlying data used to reach the conclusions drawn in the manuscript and any additional data required to replicate the reported study findings in their entirety. All PLOS journals require that the minimal data set be made fully available. For more information about our data policy, please see http://journals.plos.org/plosone/s/data-availability. Upon re-submitting your revised manuscript, please upload your study’s minimal underlying data set as either Supporting Information files or to a stable, public repository and include the relevant URLs, DOIs, or accession numbers within your revised cover letter. For a list of acceptable repositories, please see http://journals.plos.org/plosone/s/data-availability#loc-recommended-repositories . Any potentially identifying patient information must be fully anonymized.

Important: If there are ethical or legal restrictions to sharing your data publicly, please explain these restrictions in detail. Please see our guidelines for more information on what we consider unacceptable restrictions to publicly sharing data: http://journals.plos.org/plosone/s/data-availability#loc-unacceptable-data-access-restrictions . Note that it is not acceptable for the authors to be the sole named individuals responsible for ensuring data access. We will update your Data Availability statement to reflect the information you provide in your cover letter.

Thank you for bringing this to my attention and we appreciated the clarification regarding the required data set for PLOS ONE. To address this matter, we have included a 'minimal data set' in supporting information (see ‘S3_Tab’, ‘S4_Tab’, and ‘S5_Tab’). The underlying results described in my manuscript are appropriately shared in compliance with PLOS ONE guidelines. We have included the following in the supporting information files:

• Data Availability Statement: We have included a data availability statement specifying how and where people can find the data. We also explained clearly whether specific transcripts or data portions cannot be shared due to ethical considerations or participant consent restrictions.

• Description of Analysis Method: We have shared specific methods used to analyze the qualitative data, including the coding scheme, coding methods, thematic analysis, or other pertinent procedures.

• All Qualitative Codes: We have provided all the applied codes along with a description of how these codes were applied to the data.

• Description of Data Analysis Process: We have given an overview of the qualitative data analysis process, outlining steps and any iterative or cyclical processes.

• Excerpts of Transcripts: We have incorporated relevant excerpts from the qualitative interviews that played a crucial role in shaping the conclusions presented in the article. These excerpts are systematically organized according to relevant codes (see ‘S5_Tab’). 

• Member check and employee check results: Results of the member check and employee check are presented in the Supporting Information ('S4_Tab'). The raw data from the member check can be obtained by requesting it from the authors. The raw data from the employee check consists of interview transcripts and cannot be publicly shared, as employees have not explicitly consented to such sharing. However, specific data points showing which employees agreed or disagreed can be shared upon request by contacting the authors. If necessary, we can include these materials in supporting information.

6. Please ensure that you include a title page within your main document. You should list all authors and all affiliations as per our author instructions and clearly indicate the corresponding author.

Thank you for your valuable feedback. We appreciated your attention to detail. In response to your comment, we have included a title page within the main document as per your guidelines. 

Thank you. In response to your comment, we have incorporated our complete ethics statement into the 'Methods' section of the manuscript file in the additional section, titled ‘Ethical approval and consent’. This statement furnishes details on the Institutional Review Board that approved our study, providing its full name. We have also specified that informed written and verbal consent was obtained. Here is the relevant excerpt:

“Our study adhered to the stringent ethical protocol of the authors’ university. Therefore, obtaining specific ethical approval was not required, which was confirmed by the university’s Committee Ethical Affairs Faculty of Economics and Business Administration (Ghent University). Participants from the interviews, member check, and employee check provided both written and verbal informed consent, involving a detailed explanation of the study's objectives, purposes, and procedures. Participants were informed about key aspects, including the voluntary nature of their involvement, the option to withdraw at any point, the recording process with privacy safeguards, permission for data processing, and the opportunity to access the research findings. Only the first author has access to the personal information of the individual participants. Anonymity and confidentiality measures were rigorously implemented, employing code numbers, a distinct file linking personal information to these code numbers, anonymized data files, restricted access to data, and safeguards to protect any identifiable elements.” (see page 12-13, lines 295-308)

 

REVIEWER 1 

1. The study topic is interesting and deserves to have a chance as a publication to understand more about employer branding. 

Thank you for recognizing the significance of our study topic on employer branding. We are committed to providing valuable insights and contributing to the understanding of this subject. 

2. However, with the current writing, I feel that theoretical background has not been deeply analyzed. I suggest the authors to improve the section as well as the introduction part. The technical details should be expanded and clarified to ensure that readers understand exactly what the researchers studied.

Thank you for your thoughtful suggestion. In response to this suggestion, we have deepened the introduction and theoretical background. First, we have specified in the introduction the necessity of research on internal employer branding during a disruptive event and why we opt for internal communication and leadership as important touch-points. We have included and modified the following paragraphs: 

“Since the expression ‘war for talent’ suggested by McKinsey, organizations are applying different approaches to compete for talent. Literature has proposed employer branding as an effective organizational strategy to win the ‘war’ by attracting and retaining talented employees [1]. Employer branding aims to differentiate the organization, internally and externally, through its unique employment experience and attractiveness as an employer [2, 3]. To enhance attractiveness, creating and promoting a strong employer within the organization (i.e. internal employer branding) and externally promoting the employer brand through recruitment strategies (i.e. external employer branding) are crucial elements. 

Research has extensively studied the external promotion of the employer image. Internal employer branding, however, has received little attention compared to external employer branding [4]. Backhaus and Tikoo [5] described internal employer branding as a strategy that creates, promotes, and delivers a distinctive and attractive image as an employer (i.e., employer brand) to employees. To achieve an effective internal employer branding approach, the employer brand has to be present in each aspect of the organization [6]. Hence, employees should experience the internal employer brand through multiple touch-points, such as internal communication and leadership [7, 8]. These aspects, already established as important during stable times, form integral components of an effective internal employer branding strategy. Prior research suggests that a strong internal employer brand relates positively to employees’ attitudes and reduces turnover intentions [9, 10]. However, previous studies typically focused on employee perceptions, largely ignoring the crucial role that HR managers or employer brand managers play in crafting and implementing internal employer branding strategies [11]. Hence, we know little on how strategic decisions with regard to internal employer branding are made and implemented.

Furthermore, to extend our knowledge of internal employer branding, scholars like Lievens and Slaughter [12] advocate for exploring the employer brand amid unstable times, emphasizing that current literature predominantly addresses stable environments. This imperative gains significance in an era where crises, fueled by factors such as the rise of social media and accelerated media flows, occur more frequently, amplifying threats to reputations [13, 14]. During a crisis, the employer brand can be vulnerable to damage and negative influences [15]. Crises intensify the pressure on organizations to make strategic decisions. These decisions impacts the immediate well-being of employees that experience enhanced levels of frustration, uncertainty and need for information during a crisis [16]. Additionally, during crises, strategic decisions can influence employees' perceptions of the employer brand, which may prompt questions about the alignment between the organization’s stated employer brand values and the actions taken during challenging circumstances. For instance, if an organization claims to be a warm and supportive employer, will this promise hold on when things get tough and layoffs may seem necessary? Moreover, is the employer value proposition still relevant during a crisis or do certain other aspects become more important, such as creating more togetherness among employees or focusing on the operational func

---

## [Decision Letter · Decision Letter 1]

17 Mar 2024

PONE-D-23-17334R1Living the employer brand during a crisis? A qualitative study on internal employer branding in times of the Covid-19 pandemic.PLOS ONE

Dear Dr. Rys,

Thank you for submitting your revised manuscript to PLOS ONE. After careful consideration, one of the reviewers has one remaining concern for the paper to fully meet PLOS ONE’s publication criteria. Therefore, we invite you to submit a revised version of the manuscript that addresses the point raised. Please submit your revised manuscript by May 01 2024 11:59PM. If you will need more time than this to complete your revisions, please reply to this message or contact the journal office at plosone@plos.org. Please include the following items when submitting your revised manuscript:A rebuttal letter that responds to each point raised by the academic editor and reviewer(s). You should upload this letter as a separate file labeled 'Response to Reviewers'.A marked-up copy of your manuscript that highlights changes made to the original version. You should upload this as a separate file labeled 'Revised Manuscript with Track Changes'.An unmarked version of your revised paper without tracked changes. You should upload this as a separate file labeled 'Manuscript'.If applicable, we recommend that you deposit your laboratory protocols in protocols.io to enhance the reproducibility of your results. Protocols.io assigns your protocol its own identifier (DOI) so that it can be cited independently in the future. For instructions see: https://journals.plos.org/plosone/s/submission-guidelines#loc-laboratory-protocols. Additionally, PLOS ONE offers an option for publishing peer-reviewed Lab Protocol articles, which describe protocols hosted on protocols.io. Read more information on sharing protocols at https://plos.org/protocols?utm_medium=editorial-email&utm_source=authorletters&utm_campaign=protocols.

We look forward to receiving your revised manuscript.

Kind regards,

Nikolaos Georgantzis, Dr.

Academic Editor

PLOS ONE

Journal Requirements:

Reviewers' comments:

Reviewer's Responses to Questions

**Comments to the Author**

1. If the authors have adequately addressed your comments raised in a previous round of review and you feel that this manuscript is now acceptable for publication, you may indicate that here to bypass the “Comments to the Author” section, enter your conflict of interest statement in the “Confidential to Editor” section, and submit your "Accept" recommendation.

Reviewer #1: (No Response)

Reviewer #2: All comments have been addressed

2. Is the manuscript technically sound, and do the data support the conclusions?

Reviewer #1: Yes

Reviewer #2: Yes

3. Has the statistical analysis been performed appropriately and rigorously? 

Reviewer #1: N/A

Reviewer #2: N/A

4. Have the authors made all data underlying the findings in their manuscript fully available?

Reviewer #1: No

Reviewer #2: Yes

5. Is the manuscript presented in an intelligible fashion and written in standard English?

Reviewer #1: Yes

Reviewer #2: Yes

6. Review Comments to the Author

Reviewer #1: Thank you for rigorously revising the manuscript. However, I still have one question regarding the data analysis. I see that you have followed my recommendations for reanalyzing with NVivo and checking the Inter Rater Reliability. However, I found that you mentioned "we have analyzed the transcripts in NVivo. Afterwards, to ensure the reliability of our coding, we conducted an intercoder reliability assessment. Intercoder reliability assesses the agreement among multiple researchers when assigning codes to themes. Following the recommendations of O’Connor and Joffe [62] and Feng [64], we opted for Krippendorf's alpha (Kalpha) as our measure of intercoder reliability. The Kalpha, calculated in SPSS, yielded substantial agreement. This serves as validation for the coding process and demonstrates that the interpretations and assigned codes maintain consistency across multiple researchers, thereby enhancing the overall reliability and credibility of the study."

I am confused on how you said that you utilized NVivo (a qualitative analysis software) but producing the alpha numbers from SPSS (a quantitative analysis software). Is there anything that I am missing here?

Reviewer #2: I would like to express my appreciation for the authors' extra efforts to meet the reviewers' concerns. The additional data collection (6 employee member checks) and further analysis with a focus on context of warm versus competence oriented employer brand brings more nuance and richness to the conclusions. It also sharpens the contribution as internal employer branding study and not a leadership study during crisis periods. The choice on internal communication and leadership as focal touchpoints in the internal branding strategies is communicated clearly now. I can fallow this choice, but I would also highlight in limitations section (where you shortly link in line 1016 to Instrumental-Symbolic framework for other touchpoints or strategies) that based on first five interviews and prior focus in literature this choice was made on these two touchpoints with the limitation of potentially having overlooked other important actions for internal employer branding strategies during crisis, most likely I assume for more competence oriented employer brands. I would also like to invite the authors to write consistently on internal communication and leadership as touchpoints of the internal employer branding strategies to avoid or rather minimize the perception that this is a study on leadership rather than on internal employer branding choices and strategies. For example, in starting lines of Discussion line 856-861 we read “This study aimed to acquire a better understanding of how HR managers perceived and experienced internal employer branding, internal communication, and leadership during a disruptive event.” This reads too much as a study on internal employer banding and on or rather on leadership. Might seem like a detail but prevents from misinterpreting the goal of the study on strategic approaches of HR managers to bring alive the desired internal employer brand along unstable times.

In abstract it reads as if adopting warmth in the communication is the main result on internal employer branding strategy "adopting a warm communication style emerges as a facilitator in conveying the employer brand,” I would be careful as this message might conflict at first sight with the context specific approaches of more warmth and more competence oriented employer brands despite the same challenges they faced during crisis. I do believe that the finding that despite organizational/operational constraints/risks the focus and energy is devoted to the experienced internal employer brand in crisis can benefit from being emphasized more boldly in abstract (it is also as an answer to the paradox the authors describe in introduction on immediate operational problems/needs and the internal brand attention to safeguard long term reputation and retention of current employees).

An interesting finding is that employees do not per se interpret the internal branding efforts as such but rather as additional check-in time of their supervisors (see line 853) . This might point to being careful by relying on employees’ views on perceived internal employer branding efforts as for them leadership style might be at play and hence be a confounder. It is of course a challenge to distinguish a leadership style in general from embracing the internal employer brand and value proposition in how you embody your leadership as supervisor. A critical reflection on this distinction might further help set apart your study as an internal employer branding study with leadership as touchpoint and a study on leadership styles in crisis periods.

Hope these final remarks may further support the authors in having a meaningful publication in the employer branding domain.

7. PLOS authors have the option to publish the peer review history of their article (what does this mean?). If published, this will include your full peer review and any attached files.

Reviewer #1: No

Reviewer #2: **Yes: **Diane Arijs

---

## [Author Response · Author response to Decision Letter 1]

22 Apr 2024

EDITOR COMMENTS

1. Please review your reference list to ensure that it is complete and correct. If you have cited papers that have been retracted, please include the rationale for doing so in the manuscript text or remove these references and replace them with relevant current references. Any changes to the reference list should be mentioned in the rebuttal letter that accompanies your revised manuscript. If you need to cite a retracted article, indicate the article’s retracted status in the References list and also include a citation and full reference for the retraction notice.

We appreciate the chance to revise the manuscript. After a thorough review, we found no retracted papers in the reference list. However, we have made some minor adjustments to ensure accuracy, such as correcting journal names, page numbers, issue numbers, and publication dates. These adjustments are indicated in red in the document titled 'Revised Manuscript with Track Changes'. 

REVIEWER 1 COMMENTS

1. Thank you for rigorously revising the manuscript. However, I still have one question regarding the data analysis. I see that you have followed my recommendations for reanalyzing with NVivo and checking the Inter-Rater Reliability. However, I found that you mentioned "we have analyzed the transcripts in NVivo. Afterwards, to ensure the reliability of our coding, we conducted an intercoder reliability assessment. Intercoder reliability assesses the agreement among multiple researchers when assigning codes to themes. Following the recommendations of O’Connor and Joffe [62] and Feng [64], we opted for Krippendorf's alpha (Kalpha) as our measure of intercoder reliability. The Kalpha, calculated in SPSS, yielded substantial agreement. This serves as validation for the coding process and demonstrates that the interpretations and assigned codes maintain consistency across multiple researchers, thereby enhancing the overall reliability and credibility of the study." I am confused on how you said that you utilized NVivo (a qualitative analysis software) but producing the alpha numbers from SPSS (a quantitative analysis software). Is there anything that I am missing here?

Thank you for recognizing the thorough revision of the manuscript. To address this discrepancy between utilizing NVivo for qualitative analysis and generating kalpha in SPSS, we would like to clarify our methodology further. While we did analyze the interviews using NVivo for the initial coding process and thematic analysis, we encountered challenges with NVivo's functionality, including frequent crashing, which jeopardized our progress. Consequently, to ensure the reliability of our coding, we opted for a manual intercoder reliability assessment. In line with the recommendation by O'Connor and Joffe (2020), who suggested the possibility of exporting qualitative data from a qualitative platform to a statistical software package (e.g., SPSS) for reliability assessment, we followed their guidelines in assessing intercoder reliability. This involved randomizing quotes into an Excel file and engaging another researcher to assign codes based on their interpretation, guided by the theoretical framework. Subsequently, we cross-validated these assignments manually and compiled the results in an Excel spreadsheet. Afterward, we opted to utilize SPSS to calculate Kalpha as a measure of intercoder reliability. In summary, while NVivo was utilized for the initial coding, the challenges we faced necessitated a manual approach for intercoder reliability assessment, with SPSS being used for the calculation of Kalpha. 

 

REVIEWER 2 COMMENTS

1. I would like to express my appreciation for the authors' extra efforts to meet the reviewers' concerns. The additional data collection (6 employee member checks) and further analysis with a focus on context of warm versus competence-oriented employer brand brings more nuance and richness to the conclusions. It also sharpens the contribution as internal employer branding study and not a leadership study during crisis periods.

Thank you for acknowledging our efforts to address the suggestions from you and the reviewer team. We are pleased that the additional data collection and the subsequent analysis, which focuses on the warm versus competence-oriented context of employer branding, have contributed depth and nuance to our conclusions.

2. The choice on internal communication and leadership as focal touchpoints in the internal branding strategies is communicated clearly now. I can fallow this choice, but I would also highlight in limitations section (where you shortly link in line 1016 to Instrumental-Symbolic framework for other touchpoints or strategies) that based on first five interviews and prior focus in literature this choice was made on these two touchpoints with the limitation of potentially having overlooked other important actions for internal employer branding strategies during crisis, most likely I assume for more competence oriented employer brands. I would also like to invite the authors to write consistently on internal communication and leadership as touchpoints of the internal employer branding strategies to avoid or rather minimize the perception that this is a study on leadership rather than on internal employer branding choices and strategies. For example, in starting lines of Discussion line 856-861 we read “This study aimed to acquire a better understanding of how HR managers perceived and experienced internal employer branding, internal communication, and leadership during a disruptive event.” This reads too much as a study on internal employer banding and on or rather on leadership. Might seem like a detail but prevents from misinterpreting the goal of the study on strategic approaches of HR managers to bring alive the desired internal employer brand along unstable times.

Thank you for your feedback. We have adjusted the potential oversight of other important actions for internal employer branding strategies during crises. We have included following sentences in the manuscript: 

“Different theoretical frameworks, such as Instrumental-Symbolic Framework [27], could be used to incorporate a deeper understanding of how crisis events impact diverse employer brand orientations. Based on initial interviews and considering past literature, the decision was made to prioritize internal communication and leadership as touchpoints. However, we may have overlooked other important actions for internal employer strategies during times of crisis. As such, future research could also delve into (other) mechanisms or touchpoints through which organizations strategically align their internal branding efforts in response to crises, shedding light on the dynamic interplay between external uncertainties and internal organizational dynamics.” (See page 42, lines 1024-1036)

Furthermore, we have taken care to consistently emphasize internal communication and leadership as touchpoints of the internal employer branding strategies, thereby minimizing the perception that our study primarily focuses on leadership rather than on internal employer branding choices and strategies. For instance, we have revised the introductory lines of the discussion section (lines 859-861) to better align with the overarching goal of our study, which is to understand the strategic approaches of HR managers in nurturing the desired internal employer brand during times of instability.

3. In abstract it reads as if adopting warmth in the communication is the main result on internal employer branding strategy "adopting a warm communication style emerges as a facilitator in conveying the employer brand,” I would be careful as this message might conflict at first sight with the context specific approaches of more warmth and more competence-oriented employer brands despite the same challenges they faced during crisis. I do believe that the finding that despite organizational/operational constraints/risks the focus and energy is devoted to the experienced internal employer brand in crisis can benefit from being emphasized more boldly in abstract (it is also as an answer to the paradox the authors describe in introduction on immediate operational problems/needs and the internal brand attention to safeguard long term reputation and retention of current employees).

Thank you for your insightful feedback. We appreciate your attention to the clarity of our abstract. We understand your concern regarding the potential misinterpretation of our main result as solely focusing on warmth in communication, which may conflict with the diverse approaches employed by different employer brands, particularly those emphasizing competence. As such, we have revised the abstract. Following paragraph was altered in the manuscript: 

“Employer branding has emerged as a strategic imperative in the quest for talent. However, existing research has predominantly explored stable periods, overlooking the possible transformative impact of crises and the crucial role that HR managers play in crafting internal employer branding strategies. As such, this research addresses this by scrutinizing internal employer branding during the COVID-19 pandemic. Conducting in-depth interviews with 37 Belgian HR managers, we delve into the perceived challenges and opportunities that the COVID-19 crisis presented with respect to internal employer branding and its touchpoints—internal communication and leadership. A subsequent member and employee check with six HR managers and six employees validated our findings. The results unveiled organizations' heightened concern for employer branding during crises, emphasizing the strategic reflection invested. Remarkably, despite facing organizational/operational constraints/risks imposed by the crisis, the attention and efforts remain steadfastly centered on the experienced internal employer brand in crisis situations. Additionally, a contextual analysis suggests that various employer brand types face similar challenges in crises, however, the employer brand serves as a defining factor that shapes how an organization responds to both external uncertainties and internal dynamics brought about by the crisis. This study contributes to a nuanced understanding of internal employer branding dynamics during crises, shedding light on the strategic considerations of HR managers.” (See page 2, lines 26-43)

4. An interesting finding is that employees do not per se interpret the internal branding efforts as such but rather as additional check-in time of their supervisors (see line 853). This might point to being careful by relying on employees’ views on perceived internal employer branding efforts as for them leadership style might be at play and hence be a confounder. It is of course a challenge to distinguish a leadership style in general from embracing the internal employer brand and value proposition in how you embody your leadership as supervisor. A critical reflection on this distinction might further help set apart your study as an internal employer branding study with leadership as touchpoint and a study on leadership styles in crisis periods.

Thank you for highlighting this observation. We acknowledge that employees may interpret internal branding efforts differently than intended, perceiving them as additional interactions with their supervisors rather than specific employer branding initiatives. This insight underscores the importance of considering leadership style as a potential confounding factor in our study. We have taken your feedback into account and revised our paragraph, accordingly, reflecting on the distinction between internal employer branding and leadership styles during crisis periods. As such, we included in our manuscript the following paragraphs in the discussion: 

“Despite HR managers and the literature emphasizing the importance of supervisors in transferring the employer brand to employees, the participants in the employee check did not particularly perceive this. They rather perceived this as additional check-in time with their supervisor. This could suggest the need for caution when depending on employees' perspectives regarding employer branding efforts, as additional factors like leadership styles may play a role in how employees assess internal employer branding initiatives. Moreover, this incongruence between HR expectations and employee perceptions warrants further investigation. Future investigation may be required to explore the precise factors underlying this discrepancy, which could potentially stem from leadership style or the dynamics of supervisor-employee interactions. Investigating how various leadership styles, such as authoritarian, democratic, or coaching leadership, influence employees' interpretations of internal branding efforts is essential. For example, under a coaching supervisor, frequent check-ins can reinforce the perception of internal employer branding efforts by fostering supportive and collaborative interactions especially during challenging periods. Conversely, under an authoritarian supervisor, the intensified control and directive nature of conversations may play a role in how employees perceive the internal employer brand initiatives. Understanding these dynamics can provide valuable insights into optimizing the role of supervisors in internal employer branding and bridging the gap between managerial expectations and employee experiences.” (see page 43, lines 1028-1046) 

5. Hope these final remarks may further support the authors in having a meaningful publication in the employer branding domain.

Thank you for your encouraging final remarks. We appreciate your support in ensuring that our publication in the employer branding domain is meaningful.

---

## [Editor Report · Decision Letter 2]

24 Apr 2024

Living the employer brand during a crisis? A qualitative study on internal employer branding in times of the Covid-19 pandemic.

PONE-D-23-17334R2

Dear Dr. Rys,

Following a very constructive process with useful reviewer comments and suggestions to which your team has reacted in a remarkable way, I am very pleased to inform you that your manuscript is suitable for publication and will be formally accepted for publication once it meets all outstanding technical requirements.

Thank you for your contribution to PLOS.

Kind regards,

Prof. Nikolaos Georgantzis

Academic Editor

PLOS ONE

---

## [Editor Report · Acceptance letter]

30 Apr 2024

PONE-D-23-17334R2 

PLOS ONE

Dear Dr. Rys, 

I'm pleased to inform you that your manuscript has been deemed suitable for publication in PLOS ONE. Congratulations! Your manuscript is now being handed over to our production team.

Kind regards, 

on behalf of

Prof. Nikolaos Georgantzis 

Academic Editor

PLOS ONE